# Improving Calibration through the Relationship with Adversarial Robustness

**Yao Qin**   **Xuezhi Wang**   **Alex Beutel**   **Ed H. Chi**
Google Research
{yaoqin, xuezhiw, alexbeutel, edchi}@google.com

## Abstract

Neural networks lack *adversarial robustness*, i.e., they are vulnerable to adversarial examples that through small perturbations to inputs cause incorrect predictions. Further, trust is undermined when models give *miscalibrated* predictions, i.e., the predicted probability is not a good indicator of how much we should trust our model. In this paper, we study the connection between adversarial robustness and calibration and find that the inputs for which the model is sensitive to small perturbations (are easily attacked) are more likely to have poorly calibrated predictions. Based on this insight, we examine if calibration can be improved by addressing those adversarially unrobust inputs. To this end, we propose Adversarial Robustness based Adaptive Label Smoothing (`AR-AdaLS`) that integrates the correlations of adversarial robustness and calibration into training by adaptively softening labels for an example based on how easily it can be attacked by an adversary. We find that our method, taking the adversarial robustness of the in-distribution data into consideration, leads to better calibration over the model even under distributional shifts. In addition, `AR-AdaLS` can also be applied to an ensemble model to further improve model calibration.

## 1 Introduction

The robustness of machine learning algorithms is becoming increasingly important as ML systems are being used in higher-stakes applications. In one line of research, neural networks are shown to lack *adversarial robustness* – small perturbations to the input can successfully fool classifiers into making incorrect predictions (Szegedy et al., 2014; Goodfellow et al., 2014; Carlini & Wagner, 2017b; Madry et al., 2017; Qin et al., 2020b). In largely separate lines of work, researchers have studied uncertainty in model's predictions. For example, models are often *miscalibrated* where the predicted confidence is not indicative of the true likelihood of the model being correct (Guo et al., 2017; Thulasidasan et al., 2019; Lakshminarayanan et al., 2017; Wen et al., 2020; Kull et al., 2019). The calibration issue is exacerbated when models are asked to make predictions on data different from the training distribution (Snoek et al., 2019), which becomes an issue in practical settings where it is important that we can trust model predictions under distributional shift.

Despite robustness, in all its forms, being a popular area of research, the *relationship* between these perspectives has not been extensively explored previously. In this paper, we study the correlation between adversarial robustness and calibration. We discover that input data points that are sensitive to small adversarial perturbations (are easily attacked) are more likely to have poorly calibrated predictions. This holds true on a number of network architectures for classification and on all the datasets that we consider: CIFAR-10 (Krizhevsky, 2009), CIFAR-100 (Krizhevsky, 2009) and ImageNet (Russakovsky et al., 2015). This suggests that the miscalibrated predictions on those adversarially unrobust data points greatly degrades the performance of model calibration. Based on this insight, we hypothesize and study if calibration can be improved by giving different supervision to the model depending on adversarial robustness of each training data.

35th Conference on Neural Information Processing Systems (NeurIPS 2021).

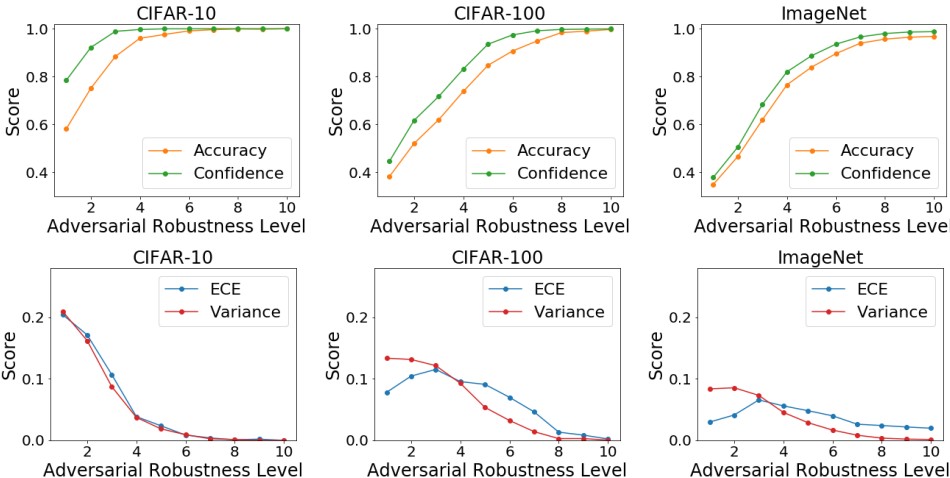

Figure 1: Inputs that are adversarially unrobust are more likely to have poorly calibrated and unstable predictions on CIFAR-10, CIFAR-100 and ImageNet. **Top:** Accuracy and confidence of the predicted class. **Bottom:** ECE (lower is better) and variance (lower is better) in each adversarial robustness subset. Higher adversarial robustness level means the input are more adversarially robust (require larger adversarial perturbations to fool the classifier into wrong predictions).

To this end, we propose **A**dversarial **R**obustness based **Ada**ptive **L**abel **S**moothing (`AR-AdaLS`) to integrate the correlations between adversarial robustness and calibration into training. Specifically, `AR-AdaLS` adaptively smooths the training labels conditioned on how vulnerable an input is to adversarial attacks. Our method improves label smoothing (Szegedy et al., 2014) by explicitly teaching the model to differentiate the training data according to their adversarial robustness and then adaptively smooth their labels. By giving different supervision to the training data, our method leads to better calibration over the model without an increase of latency during inference. In addition, since adversarially unrobust data points can be considered as outliers of the underlying data distribution (Carlini et al., 2019), our method can even greatly improve model calibration on held-out shifted data. Further, we propose "`AR-AdaLS of Ensemble`" to combine our `AR-AdaLS` and deep ensembles (Lakshminarayanan et al., 2017; Snoek et al., 2019), to further improve the calibration performance under distributional shift. Last, we find an additional benefit of `AR-AdaLS` is improving model stability (i.e., decreasing variance over multiple independent runs), which is valuable in practical applications where changes in predictions across runs (churn) is problematic.

In summary, our main contributions are as follows:

- **Relationship among Robustness Metrics:** We find a significant correlation between adversarial robustness and calibration: inputs that are unrobust to adversarial attacks are more likely to have poorly calibrated predictions.

- **Algorithm:** We hypothesize that training a model with different supervision based on adversarial robustness of each input will make the model better calibrated. To this end, we propose `AR-AdaLS` to automatically learn how much to soften the labels of training data based on their adversarial robustness. Further, we introduce "`AR-AdaLS of Ensemble`" to show how to apply `AR-AdaLS` to an ensemble model.

- **Experimental Analysis:** On CIFAR-10, CIFAR-100 and ImageNet, we find that `AR-AdaLS` is more effective than previous label smoothing methods in improving calibration, particularly for shifted data. Further, we find that while ensembling can be beneficial, applying `AR-AdaLS` to adaptively calibrate ensembles offers further improvements over calibration.

## 2 Related Work

**Uncertainty estimates**   How to better estimate a model's predictive uncertainty is an important research topic, since many models with a focus on accuracy may fall short in predictive uncertainty.

A popular way to improve a model's predictive uncertainty is to make the model well-calibrated, e.g., post-hoc calibration by temperature scaling (Guo et al., 2017), and multi-class Dirichlet calibration (Kull et al., 2019). In addition, Bayesian neural networks, through learning a posterior distribution over network parameters, can also be used to quantify a model's predictive uncertainty, e.g., Graves (2011); Blundell et al. (2015); Welling & Teh (2011). Dropout-based variational inference (Gal & Ghahramani, 2016; Kingma et al., 2015) can help DNN models make less over-confident predictions and be better calibrated. Recently, mixup training (Zhang et al., 2018) has been shown to improve both models' generalization and calibration (Thulasidasan et al., 2019), by preventing the model from being over-confident in its predictions. Despite the success of improving uncertainty estimates over in-distribution data, Snoek et al. (2019) argue that it does not usually translate to a better performance on data that shift from the training distribution. Among all the methods evaluated by Snoek et al. (2019) under distributional shift, ensemble of deep neural networks (Lakshminarayanan et al., 2017), is shown to be most robust to dataset shift, producing the best uncertainty estimates.

**Adversarial robustness**    On the other hand, machine learning models are known to be brittle (Xin et al., 2017) and vulnerable to adversarial examples (Athalye et al., 2018; Carlini & Wagner, 2017a,b; He et al., 2018; Qin et al., 2020a). Many defenses have been proposed to improve model's adversarial robustness (Song et al., 2017; Yang et al., 2019; Goodfellow et al., 2018), however are further attacked by more advanced defense-aware attacks (Carlini & Wagner, 2017b; Athalye et al., 2018). Recently, Carlini et al. (2019); Stock & Cissé (2018) define adversarial robustness as the minimum distance in the input domain required to change the model's output prediction by constructing an adversarial attack. The most recent work that is close to ours, Carlini et al. (2019), makes the interesting observation that easily attackable data are often outliers in the underlying data distribution and then use adversarial robustness to determine an improved ordering for curriculum learning. Our work, instead, explores the relationship between adversarial robustness and calibration. In addition, we use adversarial robustness as an indicator to adaptively smooth the training labels to improve model calibration.

**Label smoothing**    Label smoothing is originally proposed in Szegedy et al. (2016) and is shown to be effective in improving the quality of uncertainty estimates in Müller et al. (2019); Thulasidasan et al. (2019). Instead of minimizing the cross-entropy loss between the predicted probability $\hat{p}$ and the one-hot label $p$, label smoothing minimizes the cross-entropy between the predicted probability and a softened label $\widetilde{p} = p(1 - \epsilon) + \frac{\epsilon}{Z}$, where $Z$ is the number of classes in the dataset and $\epsilon$ is a hyperparameter which controls the degree of the smoothing effect. Our work makes label smoothing adaptive and incorporates the correlation with adversarial robustness to further improve calibration.

## 3    Correlations between Adversarial Robustness and Calibration

To explore the relationship between adversarial robustness and calibration, we first introduce the metrics to evaluate each of them (arrows indicate which direction is better).

**Adversarial robustness** ↑    Adversarial robustness measures the minimum distance in the input domain required to change the model's output prediction by constructing an adversarial attack (Carlini et al., 2019; Stock & Cissé, 2018). Specifically, given an input $x$ and a classifier $f(\cdot)$ that predicts the class for the input, the adversarial robustness is defined as the minimum adversarial perturbation $\delta$ that enables $f(x + \delta) \neq f(x)$. Following the work (Carlini et al., 2019), we construct the $\ell_2$ based CW attack (Carlini & Wagner, 2017b) and then use the $\ell_2$ norm of the adversarial perturbation $\|\delta\|_2$ to measure the distance to the decision boundary. Therefore, a more adversarially robust input requires a larger adversarial perturbation to change the model's prediction.

**Expected calibration error** ↓    Model calibration measures the alignment between the predicted probability and the accuracy. Well calibrated predictions convey the information about how much we should trust a model's prediction. We follow the widely used expected calibration error (**ECE**) to measure the calibration performance of a network (Guo et al., 2017; Snoek et al., 2019). To compute the ECE, we need to first divide all the data into $K$ buckets sorted by their predicted probability (confidence) of the predicted class. Let $B_k$ represent the set of data in the $k$-th confidence bucket. Then the accuracy and the confidence of $B_k$ are defined as $\text{acc}(B_k) = \frac{1}{|B_k|} \sum_{i \in B_k} \mathbf{1}(\hat{y}_i = y_i)$ and $\text{conf}(B_k) = \frac{1}{|B_k|} \sum_{i \in B_k} \hat{p}_i^{\hat{y}_i}$, where $\hat{y}$ and $y$ represent the predicted class and the true

Table 1: Network architecture and accuracy used for each dataset.

| Dataset | CIFAR-10 | CIFAR-100 | ImageNet |
|---|---|---|---|
| Network | ResNet-29 | WRN-28-10 | ResNet-101 |
| Accuracy | 91.4% | 79.2% | 77.7% |

class respectively, and $\hat{p}^{\hat{y}}$ is the predicted probability of $\hat{y}$. The ECE is then defined as $\text{ECE} = \sum_{k=1}^{K} \frac{|B_k|}{N} |\text{acc}(B_k) - \text{conf}(B_k)|$, where $N$ is the number of data points.

## 3.1 Correlations

Based on the evaluation metrics, we can see that adversarial robustness and calibration are measuring quite different properties: the adversarial robustness measures the property of the data by computing the adversarial perturbation $\delta$ from the *input domain*, while the calibration metric measures the properties of the model's predicted probability in the *output space*. Although adversarial robustness and calibration are conceptually different, they are both connected to the decision boundary. Specifically, adversarial robustness can be used to measure the distance to the decision boundary: if a data point is adversarially unrobust, i.e., easy to find a small input perturbation to fool the classifier into wrong classifications, then this data point is close to the decision boundary. Meanwhile, models should have relatively less confident predictions on data points close to the decision boundary. However, as pointed out by (Guo et al., 2017; Snoek et al., 2019), existing deep neural networks are frequently over-confident, i.e., having predictions with high confidence even whey they should be uncertain. Taking these two together, we hypothesize if *examples that can be easily attacked by adversarial examples are also poorly calibrated*.

To test this, we perform experiments on the clean test set across three datasets: CIFAR-10 (Krizhevsky, 2009), CIFAR-100 (Krizhevsky, 2009) and ImageNet (Russakovsky et al., 2015) with different networks, whose architecture and accuracy are shown in Table 1. We refer to these models as "Vanilla" for each dataset in the following discussion. The details for training each vanilla network are included in Appendix A.

To explore the relationship between adversarial robustness and calibration, we start with the relationship between adversarial robustness and confidence together with accuracy. Specifically, we rank the input data according to their adversarial robustness and then divide the dataset into $R$ equally-sized subsets ($R = 10$ used in this paper). For each adversarial robustness subset, we compute the accuracy and the average confidence score of the predicted class. As shown in the first row in Figure 1, we can clearly see that both accuracy and confidence increase with the adversarial robustness of the input data, and confidence is consistently higher than accuracy in each adversarial robustness subset across three datasets. This indicates that although vanilla classification models achieve the state-of-the-art accuracy, they tend to give over-confident predictions, especially for those adversarially unrobust data points.

Taking one step further, we particularly compute the expected calibration error (ECE) in each adversarial robustness subset, shown in the bottom row of Figure 1. In general, we find that data points falling into lower adversarial robustness levels are more likely to be over-confident and less well calibrated (larger ECE). For those adversarially robust examples, there is a better alignment between the model's predicted confidence and accuracy, and the ECE over those examples is close to 0. This nicely validates our hypothesis: inputs that are adversarially unrobust are more likely to have poorly calibrated predictions. On larger-scale ImageNet, while we still see the general trend holds, the least adversarially robust examples are relatively well calibrated. We hypothesize this may be due to larger training data and less overfitting.

Furthermore, we also find an interesting correlation between adversarial robustness and model stability, which is measured by the variance of the predicted probability across $M$ independent runs (e.g., $M = 5$). The variance is computed as $\sigma^2 = \frac{1}{M-1} \frac{1}{N} \sum_{m=1}^{M} \sum_{i=1}^{N} (\hat{p}_{m,i} - \bar{p}_i)^2$, where $\hat{p}_{m,i}$ is the $m$-th model's predicted probability of the $i$-th data and $\bar{p}_i = \frac{1}{M} \sum_{m=1}^{M} \hat{p}_{m,i}$ is the average predicted probability over $M$ runs. As shown in the bottom row of Figure 1, we see that those adversarially unrobust examples tend to have a much higher variance across all three datasets. This indicates that inputs that are unrobust to adversarial attacks are more likely to have unstable predictions.

**Algorithm 1** Training procedure for `AR-AdaLS`

---

**Input:** number of classes $Z$, number of training epochs $T$, number of adversarial robustness subset $R$, learning rate of adaptive label smoothing $\alpha$.

For each adversarial robustness training subset, we initialize the soft label as the one-hot label $\widetilde{p}_{r,t} = p_r$, where the initial soft label for the correct class $\widetilde{p}_{r,t}^{z=y} = 1$.

**for** $t = 1$ **to** $T$ **do**

    Minimize cross-entropy loss between soft label and predicted probability $\frac{1}{R}\sum_r^R \mathcal{L}(\widetilde{p}_{r,t}, \hat{p}_{r,t})$

    **for** $r = 1$ **to** $R$ **do**

        Update $\widetilde{p}_{r,t+1}^{z=y} \leftarrow \widetilde{p}_{r,t}^{z=y} - \alpha \cdot \{\mathrm{conf}(S_r^{val})_t - \mathrm{acc}(S_r^{val})_t\}$         ▷ Eqn. (3)

        Clip $\widetilde{p}_{r,t+1}^{z=y}$ to be within $(\frac{1}{Z}, 1]$

        Update $\epsilon_{r,t+1} \leftarrow (\widetilde{p}_{r,t+1}^{z=y} - 1) \cdot \frac{Z}{1-Z}$         ▷ Eqn. (4)

        Update $\widetilde{p}_{r,t+1} \leftarrow p_r(1 - \epsilon_{r,t+1}) + \frac{\epsilon_{r,t+1}}{Z}$         ▷ Eqn. (1)

    **end for**

**end for**

---

Taking all together, these empirical results nicely build a connection between very different concepts. In particular, adversarial robustness is measured over the input domain while both calibration and stability are measured over the output space. Given the strong empirical connection, we now ask: *can we improve model calibration and stability by targeting adversarially unrobust examples?*

## 4 Method

Based on the correlation between adversarial robustness and calibration, we hypothesize and study if calibration can be improved by giving different supervision to the model depending on the adversarial robustness of training data. To this end, we propose a method named **A**dversarial **R**obustness based **Ada**ptive **L**abel **S**moothing (`AR-AdaLS`), which performs label smoothing at different degrees to the training data based on their adversarial robustness. Specifically, we sort and divide the training data into $R$ small subsets with equal size according to their adversarial robustness[1] and then use $\epsilon_r$ to soften the labels in each training subset $S_r^{train}$. The soft labels can be formulated as:

$$\widetilde{p}_r = p_r(1 - \epsilon_r) + \frac{\epsilon_r}{Z}, \tag{1}$$

where $p_r$ stands for the one-hot vector, e.g., $p_r^{z=y} = 1$ for the correct class and $p_r^{z \neq y} = 0$ for the others, and $Z$ is the number of classes in the dataset. The parameter $\epsilon_r$ controls the degree of smoothing effect and allows for different levels of smoothing in each adversarial robustness subset. Generally, a relatively larger $\epsilon_r$ is desirable for lower adversarial robustness levels such that the model learns to make a lower confident prediction. Instead of empirically setting the parameter $\epsilon_r$ in each adversarial robustness subset, we allow it to be adaptively updated according to the calibration performance on the validation set (discussed in Section 4.1). In this way, we explicitly train a network with different supervision based on the adversarial robustness of training data.

There are two options to obtain the adversarial robustness. One is "on the fly": to keep creating the adversarial attacks during training, which provides precise adversarial robustness ranking but at the cost of great computing time. The other is to "pre-compute" the adversarial robustness by attacking a vanilla model with the same network architecture but trained with one-hot labels. This is more efficient but at the sacrifice of the precision of adversarial robustness ranking. In practice, we find that it is sufficient to make the network differentiate the adversarially robust and unrobust data with the pre-computed adversarial robustness (see more discussion in Section 5.6). Therefore, all experiments related to "`AR-AdaLS`" without further specification are based on pre-computed adversarial robustness for efficiency.

### 4.1 Adaptive learning mechanism

To find the best hyperparameter $\epsilon$ for label smoothing, previous methods (Szegedy et al., 2016; Thulasidasan et al., 2019) sweep $\epsilon$ in a range and choose the one that has the best validation

---

[1]Note, predicted confidence is not a good indicator for splitting the training dataset as the model can easily overfit to the training data and their predicted confidence are all close to 100%.

performance. However, in our setting, the number of combinations of $\epsilon_r$ increases exponentially with the number of adversarial robustness subsets $R$. To this end, we propose an adaptive learning mechanism to automatically learn the parameter $\epsilon_r$ in each adversarial robustness subset. The overall training procedure is summarized in Algorithm 1.

First, we denote the soft label for the correct class in the $r$-th adversarial robustness subset as $\widetilde{p}_r^{z=y}$. According to Eqn. (1), we can derive:

$$\widetilde{p}_r^{z=y} = 1 - \epsilon_r + \frac{\epsilon_r}{Z}. \tag{2}$$

Since well-calibrated predicted probability should be aligned with the empirical accuracy, we use the calibration performance in the *validation set* to help update $\widetilde{p}_r^{z=y}$ for the *training data*. Specifically, we first rank the adversarial robustness of the validation data and split the validation set into $R$ equally-sized subsets. Then, we use the difference between confidence and accuracy in the $r$-th adversarial robustness validation subset $\mathrm{conf}(S_r^{val}) - \mathrm{acc}(S_r^{val})$ to update the soft label for the correct class of training data in the $r$-th adversarial robustness training subset $S_r^{train}$,

$$\widetilde{p}_{r,t+1}^{z=y} = \widetilde{p}_{r,t}^{z=y} - \alpha \cdot \{\mathrm{conf}(S_r^{val})_t - \mathrm{acc}(S_r^{val})_t\}, \tag{3}$$

where $\widetilde{p}_{r,t}^{z=y}$ is the soft label of the correct class in the $r$-th adversarial robustness training subset at time step $t$. The accuracy and the confidence of $S_r^{val}$ are defined as $\mathrm{acc}(S_r^{val}) = \frac{1}{|S_r^{val}|}\sum_{i \in S_r^{val}} \mathbf{1}(\hat{y}_i = y_i)$ and $\mathrm{conf}(S_r^{val}) = \frac{1}{|S_r^{val}|}\sum_{i \in S_r^{val}} \hat{p}_i^{z=\hat{y}_i}$, where $\hat{y}$ and $y$ is the predicted class and the true class respectively, $\hat{p}^{z=\hat{y}}$ denotes the the predicted probability of the predicted class. The hyperparameter $\alpha > 0$ plays a role as a learning rate to update the soft label $\widetilde{p}_{r,t}^{z=y}$ based on the difference between the predicted confidence and accuracy in the validation set. Intuitively, if we assign a large $\widetilde{p}_r^{z=y}$ to training data, then the network tends to make a high confident prediction and vice versa. Therefore, if the confidence is greater than the accuracy ($\mathrm{conf}(S_r^{val}) > \mathrm{acc}(S_r^{val})$) in the validation set, we should reduce $\widetilde{p}_r^{z=y}$ to teach the network to be less confident. Otherwise, we should increase $\widetilde{p}_r^{z=y}$. In addition, we also need to constrain $\widetilde{p}_r^{z=y}$ to be within $(\frac{1}{Z}, 1]$ after each update as it stands for the true probability of the correct class, where $Z$ is the number of classes in the dataset.

For a given $\widetilde{p}_r^{z=y}$, we can easily obtain $\epsilon_r$ by reversing Eqn. (2),

$$\epsilon_r = (\widetilde{p}_r^{z=y} - 1) \cdot \frac{Z}{1 - Z}, \tag{4}$$

and the soft labels for all the classes $\widetilde{p}_r$ can be computed according to Eqn. (1). We update the soft labels after each training epoch in our experiments.

Note that this adaptive learning mechanism can be easily applied to standard label smoothing without adversarial robustness slicing ($R = 1$). In this case, we can replace sweeping the hyperparameter $\epsilon$ with this adaptive learning method, named as "**Ada**ptive **L**abel **S**moothing" (`AdaLS`). Our proposed `AdaLS` and `AR-AdaLS` do not increase the inference time: we test `AdaLS` and `AR-AdaLS` exactly the same as a vanilla model.

## 5 Experiments

**Datasets** We test our method on three datasets CIFAR-10, CIFAR-100 and ImageNet. In addition, we also report performance on the shifted datasets: CIFAR-10-C, CIFAR-100-C and ImageNet-C (Hendrycks & Dietterich, 2019), where there are different types of corruptions (19 types for CIFAR-10, 17 types for CIFAR-100 and 15 types for ImageNet), e.g., noise, blur, weather and digital categories that are frequently encountered in natural images. Each type of corruption has five levels of shift intensity, with higher levels having more corruption.

### 5.1 How does AR-AdaLS work?

To have a deeper understanding of how `AR-AdaLS` works, in Figure 2 we visualize the effect of label smoothing (LS) and our `AR-AdaLS`. Comparing Figure 2 (a) and (b), `AR-AdaLS` is better at calibrating the data than label smoothing, especially on the adversarially unrobust examples (lower adversarial robustness level). Further, we show plots of ECE and variance in Figure 2 (c) and (d). Both label smoothing and `AR-AdaLS` improve model calibration and stability over vanilla model and `AR-AdaLS` has the best performance among three methods. This suggests that `AR-AdaLS` is better at improving calibration and stability in adversarially unrobust regions, not just on average.

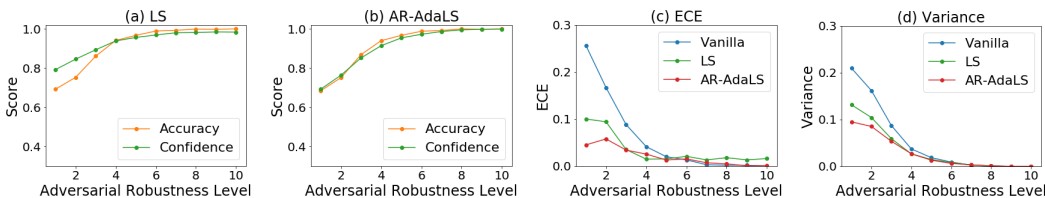

Figure 2: Comparison between LS and our `AR-AdaLS` on the clean test set of CIFAR-10. **(a)** and **(b)**: Accuracy and confidence score of the predicted class in each adversarial robustness subset. **(c)** and **(d)**: ECE and variance score of Vanilla, LS and `AR-AdaLS`.

Table 2: ECE ($\times 10^{-2}$) on CIFAR-10 and CIFAR-100. `AR-AdaLS` improves calibration and is only rivaled by domain-knowledge based data augmentation or larger ensemble models on CIFAR-100. Adversarial robustness is generated on-the-fly for `AR-AdaLS`. All the single-model based results are generated over four independent runs with random initialization.

| Method | CIFAR-10 | CIFAR-100 | Method | CIFAR-10 | CIFAR-100 |
|---|---|---|---|---|---|
| Single-model based | | | Data-augmentation based | | |
| Vanilla | 2.49±0.10 | 6.11±0.24 | mixup | 0.78±0.20 | **1.69±0.08** |
| Temperature Scaling | 0.80±0.05 | 4.26±0.07 | CCAT | 2.37±0.07 | 7.95±0.35 |
| Label Smoothing | 1.07±0.09 | 2.76±0.26 | Ensemble based | | |
| AdaLS | 1.23±0.02 | 2.65±0.31 | Mix-n-Match | 0.97 | 2.80 |
| AR-AdaLS | **0.64±0.02** | **2.27±0.16** | Ensemble of Vanilla | 0.90 | **2.21** |

## 5.2 AR-AdaLS improves calibration

**Baselines** We compare our proposed `AR-AdaLS` with the following 8 different methods: (1) Vanilla model trained with one-hot labels, (2) Temperature Scaling (Guo et al., 2017), a post-hoc calibration method where the predicted logits are divided by a temperature which is tuned on the hold-out validation set, (3) label smoothing (LS) (Szegedy et al., 2016) that softs labels by sweeping the hyperparameter $\epsilon$ (the smoothing degree) in a range to find the best hyperparameter $\epsilon$, (4) Adaptive Label Smoothing (`AdaLS`): we use our proposed adaptive learning mechanism introduced in Section 4.1 to automatically learn the hyperparameter $\epsilon$ rather than sweeping to find the best $\epsilon$. (5) mixup, which is a data augmentation technique originally proposed in (Zhang et al., 2018) and recently found to be able to improve calibration in (Thulasidasan et al., 2019), (6) Confidence-calibrated adversarial training (CCAT) (Stutz et al., 2020), a method builds on adversarial training by reducing the confidence in the labels of adversarial examples. Note that there is a significant difference between our `AR-AdaLS` and CCAT: CCAT trains a model on the generated adversarial examples to improve a model's adversarial robustness. In contrast, our `AR-AdaLS`, trained on the clean training data, is proposed to use the correlation between adversarial robustness and calibration to improve a model calibration performance. (7) "Ensemble of Vanilla" (Lakshminarayanan et al., 2017), an ensemble of $M$ vanilla models independently trained with random initialization. (8) Mix-n-Match (Zhang et al., 2020), an ensemble and compositional method proposed for calibration. All the methods are trained with the same network architecture, i.e., WRN-28-10 (Zagoruyko & Komodakis, 2016) on both CIFAR-10 and CIFAR-100, and the same training hyperparameters: e.g., learning rate, batch size, number of training epochs, for fair comparison[2]. Please refer to Appendix A for all the training details and hyperparameters.

**Results** The expected calibration error of all the methods on CIFAR-10 and CIFAR-100 are displayed in Table 2. We can clearly see that by differentiating the training data based on their adversarial robustness, `AR-AdaLS` effectively reduces the calibration error compared to other single-model based methods without significant change in accuracy (see Figure 6 in Appendix) and it is only rivaled by mixup on CIFAR-100, which uses extra domain knowledge through data augmentation. Note that `AR-AdaLS` is only trained on the clean training data without any data augmentation compared to mixup (Thulasidasan et al., 2019) and CCAT (Stutz et al., 2020).

---

[2]The result of Mix-n-Match in Table 2 is from Table 1 reported in the original work (Zhang et al., 2020), which is trained with the same network architecture, WRN-28-10.

Table 3: Mean of ECE ($\times 10^{-2}$) across 19 types of shift for CIFAR-10-C and 15 types of shift for ImageNet-C. Smaller is better. ResNet-29 is used for CIFAR-10 and ResNet-101 is used for ImageNet. The standard deviation of five independent runs for each single model is reported. The best single model and ensemble model in each shift intensity is highlighted in **bold**.

| Single-model based | | | Ensemble-based | | |
|---|---|---|---|---|---|
| Methods | CIFAR-10-C | ImageNet-C | Methods | CIFAR-10-C | ImageNet-C |
| Vanilla | 16.7±0.5 | 10.7±0.5 | Ensemble of Vanilla | 6.5 | 4.2 |
| LS | 10.1±0.4 | 8.1±0.4 | Ensemble of LS | 4.6 | 4.7 |
| AdaLS | 9.6±0.5 | 8.0±0.2 | Ensemble of AdaLS | 5.2 | 4.8 |
| AR-AdaLS | **6.4**±0.6 | **6.9**±**0.2** | Ensemble of AR-AdaLS | 5.5 | 5.1 |
| | | | AR-AdaLS of Ensemble | **4.4** | **4.0** |

## 5.3 Improve calibration on shifted dataset

Table 3 summarizes the mean calibration error (ECE) on the corrupted datasets: CIFAR-10-C and ImageNet-C (Hendrycks & Dietterich, 2019). Looking at all the *single-model* based methods, we can see that AR-AdaLS significantly outperforms other single-model based methods with the lowest ECE. Contrasting with LS and AdaLS, we see AR-AdaLS benefits greatly from the adversarial robustness slicing. As a result, our model learns to give smaller soft labels of the correct class to those adversarially unrobust training data, which can also be considered as outliers of the underlying data distribution (Carlini et al., 2019). Therefore, when tested on the shifted data that deep networks have been shown to produce pathologically over-confident predictions (Hendrycks & Dietterich, 2019), our model correctly learns to make a relatively lower-confidence prediction, resulting in a better calibration performance.

In addition, we also compare AR-AdaLS with "Ensemble of Vanilla" (Lakshminarayanan et al., 2017), which is shown to be the best model for models' calibration under distributional shift (Snoek et al., 2019). The result of Ensemble of Vanilla is an ensemble of $M = 5$ vanilla models independently trained with random initialization. We can see that AR-AdaLS achieves comparable calibration performance on CIFAR-10 and the ensemble is better under highly shifted data on ImageNet.

**Combination with deep ensembles** We further discuss the following two ways to combine AR-AdaLS with ensembles:

- Ensemble of AR-AdaLS: As in Lakshminarayanan et al. (2017), we ensemble AR-AdaLS by training multiple independent AR-AdaLS models with random initialization, and average their predictions at inference.

- AR-AdaLS of Ensemble: Instead of computing soft labels independently for each AR-AdaLS, we perform AR-AdaLS on the ensembled predictions, i.e., in Eqn (3) we compute confidence and accuracy based on the average of $M = 5$ model predictions. Each model is then supervised with the same soft labels. We will see this slight distinction in training is quite important.

As shown in Table 3, naively combining deep ensembles with AR-AdaLS (Ensemble of AR-AdaLS) could not effectively improve models' calibration (see more details in Appendix B). In contrast, AR-AdaLS of Ensemble, which adaptively adjusts smoothing to keep the ensemble models well calibrated, performs the best under distributional shift on both CIFAR-10 and ImageNet.

## 5.4 Improve model stability

Since we observe in Figure 1 that the most adversarially unrobust data points also have very unstable predictions, we test AR-AdaLS to see if it can help improve model stability, which is of great value in practice where high variance of a model is bad for churn (Milani Fard et al., 2016). In Figure 3 we can see that AR-AdaLS can effectively reduce the variance of a model compared to a vanilla model and label smoothing on CIFAR-10 and ImageNet. Please refer to Table 5 in Appendix for numerical numbers on both datasets.

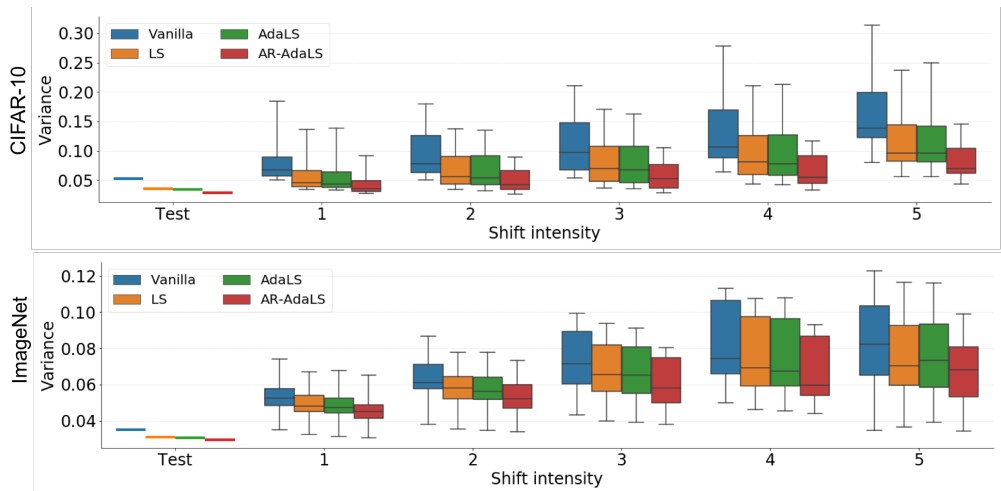

Figure 3: Variance on clean test and shifted data on CIFAR-10 and ImageNet. For each shift intensity, we show the results with a box plot summarizing the 25th, 50th, 75th quartiles across 19 shift types on CIFAR10-C and 15 shift types on ImageNet-C. The error bars indicate the $min$ and $max$ value across different shift types. ResNet-29 is used for CIFAR-10 and ResNet-101 is used for ImageNet.

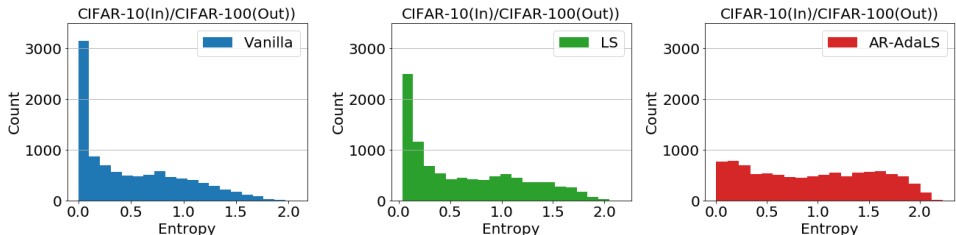

Figure 4: Histogram of predictive entropy on out-of-distribution data. Each model is trained on CIFAR-10 and tested on CIFAR-100. The network architecture is WRN-28-10 and adversarial robustness in `AR-AdaLS` is generated on-the-fly.

### 5.5 Improvements on out-of-distribution data

We further study the performance of `AR-AdaLS` when predicting on out-of-distribution (OOD) data. Following (Snoek et al., 2019), we compare the performance of Vanilla, Label Smoothing and `AR-AdaLS` by plotting the histogram of the entropy on the OOD data (higher entropy on OOD is better). As shown in Figure 4, each model is trained on CIFAR-10 dataset and then tested on CIFAR-100 dataset. We can clearly see that `AR-AdaLS` significantly reduces the number of low-entropy predictions on OOD data. In addition, using CIFAR-10/CIFAR-100 as in-distribution/out-of-distribution data, we also report the Area under the ROC curve (AUROC) of label smoothing, mixup and `AR-AdaLS`. The AUROC score of standard label smoothing and mixup is 0.832±0.005 and 0.821±0.003 respectively, whereas our `AR-AdaLS` achieves 0.885±0.003. This demonstrates the effectiveness of `AR-AdaLS` even on fully out-of-distribution data.

### 5.6 Sensitivity analysis

**Sensitivity to the number of adversarial robustness subsets**   We perform a sensitivity analysis for the number of adversarial robustness subsets $R$. Specifically, we plot the calibration error of `AR-AdaLS` with a varying $R$ on the clean CIFAR-10 and corrupted CIFAR-10-C in Figure 5. We can see that there is a significant drop in calibration error (ECE) when we increase the number of adversarial robustness subsets $R$ from 1, where $R = 1$ denotes `AdaLS`. Further, the calibration error is relatively stable when $R$ is chosen within the range [10, 16]. Thus, we choose $R = 10$ for all results shown in this paper for `AR-AdaLS`.

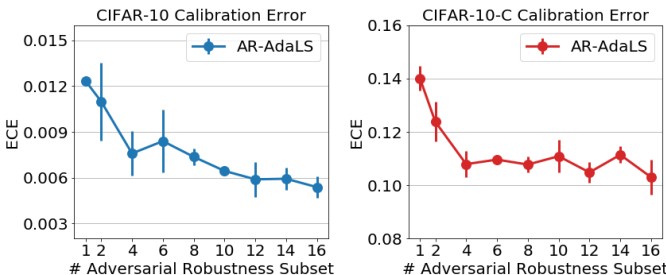

Figure 5: ECE on CIFAR-10 and CIFAR-10-C of `AR-AdaLS` with varying number of adversarial robustness subset $R$. Note that when $R = 1$, `AR-AdaLS` becomes `AdaLS`. The results are based on 4 independent runs.

Table 4: Ablation study of `AR-AdaLS` on CIFAR-100 and CIFAR100-C (corrupted). We report both accuracy ($\times 10^{-2}$) and expected calibration error ($\times 10^{-2}$), denoted by **Acc** and **ECE** for the clean test set, and **cAcc** and **cECE** for CIFAR100-C. Arrow indicates the better direction; best calibration is **bolded**.

| Method | Vanilla | Label Smoothing | Temperature Scaling | AR-AdaLS (pre-compute) | AR-AdaLS (on-the-fly) |
|---|---|---|---|---|---|
| **Acc/cAcc** ($\uparrow$) | 79.2/52.0 | 78.9/51.7 | 79.2/52.0 | 79.3/52.2 | 79.2/52.1 |
| **ECE/cECE** ($\downarrow$) | 6.1/18.2 | 2.8/16.3 | 4.3/14.0 | 2.6/14.2 | **2.3/13.2** |

**Sensitivity to the exactness of adversarial robustness** To investigate this, we study the performance of `AR-AdaLS` using adversarial robustness generated via two different ways: One is "on-the-fly": we keep creating adversarial attacks during training, which provides a more precise adversarial robustness ranking but at the cost of great computing time. The other is to "pre-compute" adversarial robustness by attacking a vanilla model that is trained with one-hot labels. This is more efficient but at the sacrifice of the precision of adversarial robustness ranking. We perform experiments on CIFAR-100 as an example to compare the performance of `AR-AdaLS` based on the adversarial robustness that is "pre-computed" or "on-the-fly". As shown in Table 4, generating adversarial robustness "on-the-fly" can further help improve the calibration performance for `AR-AdaLS` on both clean and shifted datasets, compared to pre-computing adversarial robustness. Similar patterns are observed on CIFAR-10.[3]

Therefore, we can conclude that 1) the exactness of adversarial robustness is helpful for `AR-AdaLS`, that is, more precise adversarial robustness leads to a better performance. 2) `AR-AdaLS` with an approximation of adversarial robustness (pre-computed) can already significantly improve label smoothing. Hence, all results in this paper related to "`AR-AdaLS`" without further specification are based on pre-computed adversarial robustness for efficiency. This is because our main target is to show that the idea of differentiating the training data based on their adversarial robustness is promising to improve model calibration rather than pushing the results to the best.

## 6 Conclusion

In this paper, we have explored the correlations between adversarial robustness and calibration. We find across three datasets that adversarially unrobust data points, where small adversarial perturbations to the input are able to fool the classifier into wrong predictions, are more likely to have poorly calibrated and unstable predictions. Based on this insight, we propose `AR-AdaLS` to adaptively smooth the labels of the training data based on their adversarial robustness. In our experiments we see that `AR-AdaLS` is more effective than previous label smoothing methods in improving calibration, particularly for shifted data, and can offer improvements on top of already strong ensembling methods. We believe this is an exciting new use for adversarial robustness as a means to more generally improve model trustworthiness, not just by limiting adversarial attacks but also improving calibration and stability on unexpected data. We hope this spurs further work at the intersection of these areas of research.

---

[3]We did not run on-the-fly `AR-AdaLS` for ImageNet due to the computational intensity.

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
