# A Implementation Details

## A.1 CIFAR-10

**ResNet-29** For all the experimental results on ResNet-29 v2 (He et al., 2016b), we use a batch size of 256. The network is trained with Adam optimizer (Kingma et al., 2015) for 200 epochs. The initial learning rate is $10^{-3}$ and decayed down to $10^{-4}$ after 80 epochs, $10^{-5}$ after 120 epochs, $10^{-6}$ after 160 epochs and $0.5 \times 10^{-6}$ after 180 epochs. We adapted the following data augmentation and training script at `https://keras.io/examples/cifar10_resnet/`. The training mechanism is the same for all the methods that we compare in the main paper. We randomly split the training dataset into training data of 45000 images and 5000 images as the validation set. The test set has 10000 images.

For label smoothing (LS), we sweep the hyperparameter $\epsilon$ within the range [0, 0.1] with a step size 0.01 and find that the network has the best calibration performance on the validation set when $\epsilon = 0.02$.

For Adaptive Label Smoothing (`AdaLS`), there is a hyperparameter $\alpha$ which plays a role as learning rate in the adaptive learning mechanism. We choose hyperparameter $\alpha$ based on the calibration performance on the validation set. Specifically, we run experiments with $\alpha \in \{0.005, 0.01, 0.05, 0.1\}$ and find that $\alpha = 0.05$ achieve the best calibration performance.

Similarly, for Adversarial Robustness based Adaptive Label Smoothing (`AR-AdaLS`), we choose the hyperparameter $\alpha$ from the set $\{0.005, 0.01, 0.05\}$ and empirically set $\alpha = 0.005$ which has the best calibration performance on the validation set. We update the training labels after each epoch for all the experiments related to `AR-AdaLS`, including the experiments on CIFAR-100 and ImageNet. We use the same hyperparameter $\alpha = 0.005$ without further tuning for `AR-AdaLS` of Ensemble.

All the results of ensemble models are obtained via training 5 independent models with random initializations.

**WRN-28-10** We train a Wide ResNet-28-10 v2 (Zagoruyko & Komodakis, 2016) to obtain the state-of-the-art accuracy for CIFAR-10 (e.g., Table 2 in the main text). We adapt the same training details and data augmentation at `https://github.com/google/edward2/blob/master/baselines/cifar/deterministic.py`.

For label smoothing, we sweep the hyperparameter $\epsilon$ within the range [0, 0.1] with a step size 0.01 and find that the network has the best calibration performance on the validation set when $\epsilon = 0.02$.

For `AdaLS` and `AR-AdaLS`, the hyperparameter $\alpha$ is set to be 0.005. For `AR-AdaLS` that generated with on-the-fly adversarial examples, we recompute the adversarial robustness for training and validation sets after 65, 130 epochs.

For mixup (Zhang et al., 2018; Thulasidasan et al., 2019), the mixing parameter of two images is randomly sampled from a Beta distribution Beta($\beta, \beta$) at each training iteration. We set $\beta = 0.2$ for best calibration performance on in-distribution data.

For CCAT (Stutz et al., 2020), we observe that training models with adversarial examples bounded with smaller $\ell_\infty$ norm, e.g., $||\delta||_\infty \leq 0.01$, can benefit more to the calibration with a small accuracy sacrifice on the clean data. Therefore, we train CCAT with PGD attacks bounded by $||\delta||_\infty \leq 0.01$. The step size and total iterations to generate PGD attacks is 0.025 and 10 respectively during training.

All the results of ensemble models on WRN-28-10 are obtained via training 4 independent models with random initializations.

## A.2 CIFAR-100

We train a Wide ResNet-28-10 v2 (Zagoruyko & Komodakis, 2016) to obtain the state-of-the-art accuracy for CIFAR-100. We adapt the same training details and data augmentation at `https://github.com/google/edward2/blob/master/baselines/cifar/deterministic.py`.

For label smoothing, we e sweep the hyperparameter $\epsilon$ within the range [0, 0.1] with a step size 0.01 and find that the network has the best calibration performance on the validation set when $\epsilon = 0.07$.

All the hyperparameters used for `AdaLS`, `AR-AdaLS`, mixup (Zhang et al., 2018; Thulasidasan et al., 2019) and CCAT (Stutz et al., 2020) are the same as those for CIFAR-10 with WRN-28-10.

All the results of ensemble models on WRN-28-10 are obtained via training 4 independent models with random initializations.

### A.3  ImageNet

All the experiments on ImageNet were obtained via training a ResNet-101 v1 (He et al., 2016a) following the training script at `https://github.com/google/edward2/blob/master/baselines/imagenet/deterministic.py`. The network is trained with a batch size of 128 for each TPU core with SGD optimizer for 90 epochs. The input image is normalized (divided by 255) to be within [0,1]. We randomly divide 50000 validation images into validation set with 25000 images and test set with 25000 images. Note that the same dataset and training mechanisms are used for all the methods that we compare in the main paper.

For Label Smoothing (LS), we sweep the hyperparameter $\epsilon$ within the range [0, 0.1] with a step size 0.01 and find that the best calibration performance on the validation set is achieved by setting $\epsilon = 0.02$.

For Adaptive Label Smoothing (`AdaLS`), we sweep the hyperparameter $\alpha$ in the set $\{0.005, 0.01, 0.03, 0.05, 0.1\}$ and set it to be $\alpha = 0.03$ for the best calibration performance on the validation set.

We empirically set $\alpha = 0.001$ for `AR-AdaLS` in the first 60 epochs of the training and then increase it to 0.05 for the next 30 epochs. The same hyperparameter $\alpha$ is used for `AR-AdaLS` of Ensemble without further tuning.

All the ensemble models are a combination of 5 independent models with random initializations.

### A.4  CW attacks

To compute the adversarial robustness, we construct $\ell_2$ based CW attacks (Carlini & Wagner, 2017b) following the code at `https://github.com/tensorflow/cleverhans/blob/master/cleverhans/attacks/carlini_wagner_l2.py`. Specifically, we set the binary search steps to be 3, max iterations to be 500 and learning rate to be 0.005. The generated untargeted CW attacks can achieve 100% success rate for all the datasets that we consider: CIFAR-10, CIFAR-100 and ImageNet. We set the number of adversarial robustness training subset and validation subset to be $R = 10$ respectively.

## B  Discussion of AR-AdaLS Combined with Ensembles

In Figure 6, we show ECE and accuracy of all the single-models and their corresponding ensembles on the clean test and shifted CIFAR-10 and ImageNet. At a high level, we see that all the ensemble models that we compare have similar accuracy, which are higher than single-models. `AR-AdaLS of Ensemble` performs the best across both clean test data and all intensities of shifted data in terms of calibration.

Looking more closely, some trends emerge: all of the ensemble methods perform relatively well for highly shifted data (intensity 4–5), but Ensemble of LS, Ensemble of `AdaLS`, `Ensemble of AR-AdaLS` perform much worse on less shifted and clean test data. Digging deeper, we display the confidence of the predicted class and accuracy of each single model and the corresponding ensemble models on the clean test set of CIFAR-10 and ImageNet in Figure 7. We can clearly see that the ensemble models generally increase accuracy and decrease confidence compared to a single model, which results from the disagreement of the prediction of each single model in ensembles. Therefore, naive deep ensembles can improve calibration on highly shifted data where single-model is over-confident but can harm calibration if applied to a well-calibrated single-model. This is made clearer in Figure 8: while deep ensembles make over-confident vanilla model well calibrate, it leads the well calibrated models, e.g., `AR-AdaLS`, to be under-confident. From this perspective, `AR-AdaLS of Ensemble` avoids this issue by adaptively adjusting smoothing to keep the ensembles well calibrated on both clean and shifted dataset.

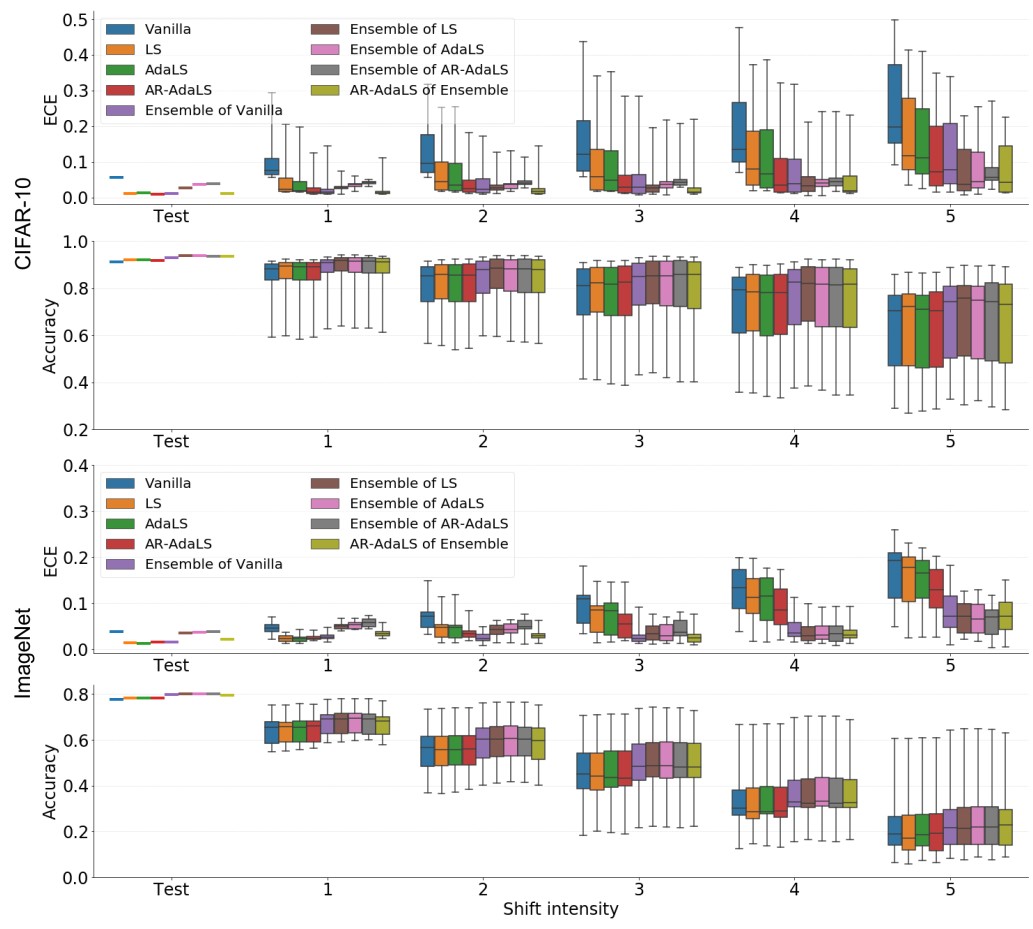

Figure 6: Comparison of ensemble models: ECE and Accuracy on both clean test data and shifted data on CIFAR-10 and ImageNet. For each intensity of shift, we show the results with a box plot summarizing the 25th, 50th, 75th quartiles across 19 types of shift on CIFAR-10-C and 15 types of shift on ImageNet-C. The error bars indicate the $min$ and $max$ value across different shift types.

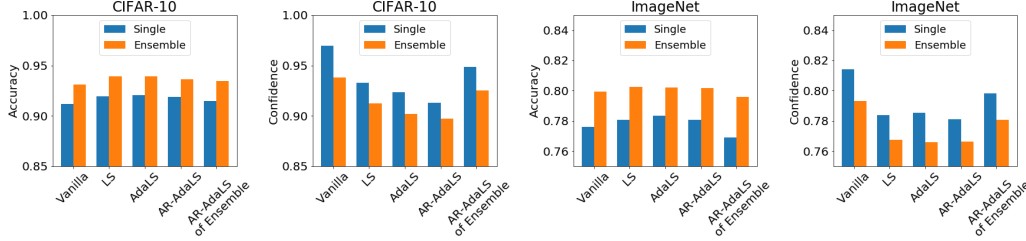

Figure 7: Comparing accuracy and confidence of the predicted class between single model and the corresponding ensemble model for each method.

Table 5: Mean of variance ($\times 10^{-2}$) across 19 types of shift for CIFAR-10-C and 15 types of shift for ImageNet-C. Best in **bold**.

| Dataset | CIFAR10-C | | | | | | ImageNet-C | | | | | |
|---|---|---|---|---|---|---|---|---|---|---|---|---|
| Shift Intensity | 1 | 2 | 3 | 4 | 5 | Mean | 1 | 2 | 3 | 4 | 5 | Mean |
| Vanilla | 7.85 | 9.69 | 11.2 | 13.1 | 16.0 | 11.6 | 5.28 | 6.39 | 7.37 | 8.23 | 8.29 | 7.11 |
| LS | 5.54 | 6.95 | 8.11 | 9.65 | 11.8 | 8.41 | 4.86 | 5.84 | 6.78 | 7.55 | 7.41 | 6.49 |
| AdaLS | 5.47 | 6.87 | 7.95 | 9.44 | 11.5 | 8.25 | 4.79 | 5.77 | 6.66 | 7.51 | 7.56 | 6.46 |
| AR-AdaLS | **4.21** | **5.06** | **5.73** | **6.66** | **8.24** | **5.98** | **4.53** | **5.49** | **6.12** | **6.76** | **6.66** | **5.91** |

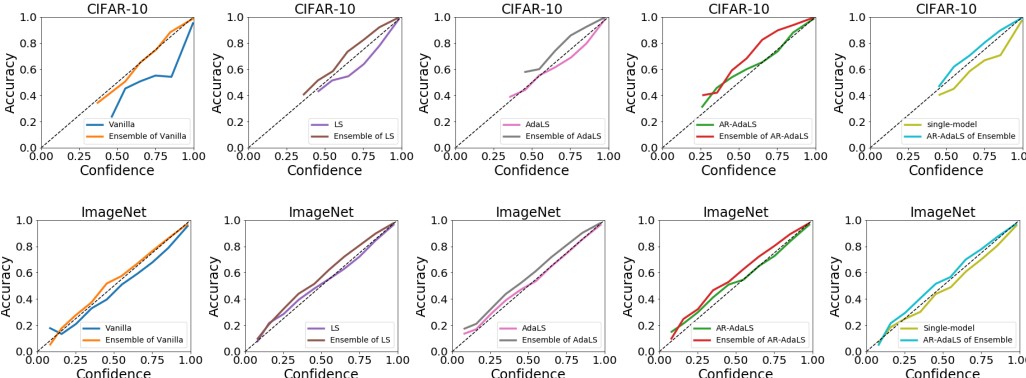

Figure 8: Reliability diagram of accuracy versus confidence of single model and ensemble model on the clean test of CIFAR-10 and ImageNet. The perfect calibrated model should be aligned with the diagonal dotted line (above is under-confident, below is over-confident).