# OpenReview forum: "Improving Calibration through the Relationship with Adversarial Robustness"
_NeurIPS.cc/2021/Conference — NeurIPS 2021 Poster_

### Official Review · Reviewer_dquZ · 2021-07-15

**Rating:** 7
**Confidence:** 3

**Summary:**

This paper studies the connection between adversarial robustness (measured as the distance to the decision boundary using some adversarial attack) and model calibration (i.e how well the predicted probability indicates how much we can truth the model predictions).

1. The authors show that there is a significant correlation between adversarial robustness and model calibration. That is, inputs that have smaller distance to the decision boundary are more likely to have poorly calibrated predictions.
2. Based on this insight, the authors propose an algorithm called AR-AdaLS to learn how much to smooth the labels of the training data based on their adversarial robustness. They also discuss how this technique can be extended to an ensemble model.
3. The authors thoroughly compare their results against many previous calibration techniques and show that AR-AdaLS results in improved performance (for the single-model based methods). The results are superior also under distribution shift. They show results on CIFAR-10, CIFAR-100 and Imagenet datasets. For results under distribution shift, they show results on CIFAR-10-C, CIFAR-100-C and Imagenet-C datasets.

**Limitations And Societal Impact:**

I see no foreseeable potential negative societal impacts of their work.

**Main Review:**

The paper studies the important problem of model calibration and provides a novel viewpoint based on the idea of using adversarial robustness (measured using distance to the decision boundary). The authors clearly discuss how the work differs from the previous related works and in which cases the performance of their proposed method is better.

The submission is technically sound. The claims are well supported through extensive experiments on well studied datasets such as CIFAR-10, CIFAR-100 and Imagenet. The results under distribution shift are also extensively compared on CIFAR-10-C, CIFAR-100-C and Imagenet-C datasets.

The paper is well written and easy to understand.

I believe the results showing link between the two seemingly different research areas of adversarial robustness and model calibration will be interesting for the research community.
The authors perform a very thorough comparison of their method against many single-model based methods and ensemble-based or data-augmentation based methods and find that their approach results in superior performance over single-model based methods. They show results on CIFAR-10, CIFAR-100 and Imagenet datasets.

For results under distribution shifts, they show results on CIFAR-10-C, CIFAR-100-C and Imagenet-C datasets and they are significantly better than the previous methods. The authors also show results on CIFAR-100 when the model is trained on CIFAR-10 and the number of low entropy predictions on OOD data is reduced.

**Time Spent Reviewing:**

6

---

> ### Author Response · Authors · 2021-08-10
> **Response to Reviewer dquZ**
>
> Thanks very much for supporting our work and we really appreciate that you acknowledge building a connection between the two seemingly different research areas of adversarial robustness and model calibration is impactful for the research community. We believe this work can spur further work at the intersection of different research areas related to robustness.

---

### Official Review · Reviewer_ceCu · 2021-07-16

**Rating:** 7
**Confidence:** 5

**Summary:**

1) This paper points out a novel finding: Examples that can be adversarially attacked may also be calibrated worse.
2) The author utilize the above finding to design the AR-AdaLS to enhance calibration.

**Main Review:**

Generally speaking, I am satisfied with the topic and the method proposed in this work. In these two years, adversarial robustness has been found to be related to many other concurrent fields: Outlier Detection[1], Domain Adaption[2], Data Augmentation[3] etc. According to my knowledge, this is the first work that points out the relationship between adversarial robustness and calibration. Therefore, I think this can complement the current study about the concept of the general robustness. The proposed method is reasonable and consistent with the finding. The experimental setup aligns with common choices in other works. In particular, I consider this work as a good extension for [4]. However, I still have a few concerns:
1) Although I do think the observation that "Examples that can be adversarially attacked may also be calibrated worse" is reasonable, I find that in this paper the authors did not provide sufficient information to properly explain this phenomena. Section 3 are basically composed of empirical findings and lack of analyses. I hope the authors can offer more insights in the rebuttal phase and perhaps provide evidences from other works, or maybe explain the phenomena based on the framework of [4].
2) In Table 3, the author combine the proposed AR-AdaLS with ensemble based method. Can AR-AdaLS be combined with the data augmentation based methods like MixUp as in Table 1?
3) Although the writing is rather clear and straightforward, there are many typos or mistakes. For instance, the appendix is submitted with the main content. Many section titles are not properly capitalized. Of course, these minor issues do not influence my general impressions on this paper. But I suggest the author to be more careful with these mistakes in the future.








[1] ENHANCING THE RELIABILITY OF OUT-OF-DISTRIBUTION IMAGE DETECTION IN NEURAL NETWORKS

[2] Adversarial Examples Improve Image Recognition

[3] Adversarial Mutual Information for Text Generation

[4] When does label smoothing help?

**Time Spent Reviewing:**

8

---

> ### Author Response · Authors · 2021-08-10
> **Response to Reviewer ceCu**
>
> Thanks very much for accepting the valuable insights from our work which builds a connection between adversarial robustness and calibration. We really appreciate that you acknowledge the importance of this work to the research community working on different concepts of machine learning robustness. We answer your questions below.
>
> Q1: Thanks very much for suggesting that we provide more explanation on the correlation between adversarial robustness and calibration. We will add a deeper discussion in the final version. To briefly discuss here, our work is inspired by [1] and [2], where adversarial robustness can be used to measure the distance to the decision boundary. Specifically, if a data point is adversarially unrobust, i.e., easy to find an input perturbation which makes the classifier change the classification, then this data point is close to the decision boundary. Meanwhile, data points that are close to the decision boundary should be uncertain, as shown in Figure 1 in [3] where the uncertainty surface is visualized.  Combining these two observations, we hypothesize that adversarially unrobust examples are more likely to be uncertain data points as they are both close to the decision boundary.
>
> In addition, as pointed out by [4, 5], existing deep neural networks are frequently over-confident, i.e., having predictions with high confidence even when they should be uncertain. This indicates that those uncertain data points are more likely to be poorly calibrated.
>
> Taking these two together, we investigate if "examples that can be adversarially attacked may also be calibrated worse" and the empirical results validate our hypothesis.
>
> Q2: Combine with data augmentation: Our method can be easily generalized to standard data augmentation techniques like RandAug [6], Augmix [7], etc, which use one-hot labels. The simplest way is to compute the adversarial robustness of the augmented data and then smooth their labels accordingly. However, for mixup, which originally involves mixing the labels of two different images, we need to be more careful about assigning the labels for the augmented images. The general insight of this work, however, holds for any data augmentation: we can smooth the labels for augmented data based on the distance to the decision boundary (we show in this work that adversarial robustness is a good indicator of this distance).
>
> Q3: Thanks very much for pointing out the issues with the appendix, section titles and other typos. We will be very careful and correct all the mistakes in the final version.
>
>
> [1] Stock, P. and Cissé, M. Convnets and imagenet beyond accuracy: Understanding mistakes and uncovering biases. ECCV 2018.
>
> [2] Carlini, Nicholas, Ulfar Erlingsson, and Nicolas Papernot. "Distribution density, tails, and outliers in machine learning: Metrics and applications." arXiv preprint arXiv:1910.13427 (2019).
>
> [3] Liu, Jeremiah Zhe, et al. "Simple and principled uncertainty estimation with deterministic deep learning via distance awareness." NeurIPS 2020.
>
> [4] Guo, C., Pleiss, G., Sun, Y., and Weinberger, K. Q. On calibration of modern neural networks. ICML 2017.
>
> [5] Snoek, J., Ovadia, Y., Fertig, E., Lakshminarayanan, B., Nowozin, S., Sculley, D., Dillon, J., Ren, 375 J., and Nado, Z. Can you trust your model’s uncertainty? evaluating predictive uncertainty under 376 dataset shift. NeurIPS 2019.
>
> [6] Cubuk, Ekin D., et al. "Randaugment: Practical automated data augmentation with a reduced search space." Proceedings of the IEEE/CVF Conference on Computer Vision and Pattern Recognition Workshops. 2020.
>
> [7] Hendrycks, Dan, et al. "Augmix: A simple data processing method to improve robustness and uncertainty." ICLR 2020.

---

> > ### Comment · Reviewer_ceCu · 2021-08-24
> > **Thanks for the response**
> >
> > Most of my concerns are solved by your response. Thanks. I decide to keep my score unchanged.

---

### Official Review · Reviewer_jdF6 · 2021-07-18

**Rating:** 6
**Confidence:** 4

**Summary:**

The authors first expose a link between robustness and expected calibration error (ECE), the less robust a data point is, the larger the ECE. They then propose to exploit this link by introducing an adaptive label smoothing method that improves the expected calibration error of less robust data points. They benchmark their new method showing better calibration metrics on standard datasets, as well on corrupted datasets and out-of-distribution data.

**Limitations And Societal Impact:**

The limitations are not addressed.

**Main Review:**

The authors show that their adaptive label smoothing improves upon standard label smoothing. They also show how the adaptive label smoothing can further be improved by splitting the data into several adversarial robustness bins. However, I have several concerns/questions.


1. Mixup and other baselines. From Tab. 2 we see that Mixup outperforms the proposed method on CIFAR-100, whereas on CIFAR-10 Mixup is competitive to the proposed method.
However, in the rest of the paper, the authors do not show results with Mixup. Also, additional baselines are missing in the evaluation on corrupted datasets and out-of-distribution data, which prevents the reader from entirely assessing the proposed method.
In my opinion, such benchmarks, and ideally both with mixup and with competitive methods to mixup, are necessary to indeed convey the message that the proposed method works well relative to existing methods.

2. While the authors use CIFAR-100 and ImageNet in Fig. 1 to show the correlation between confidence and adversarial robustness, most of the results that follow focus solely on CIFAR-10. Could the authors show the results on CIFAR-100 and ImageNet (e.g. in Fig.2)? In my opinion, such results are necessary to indeed conclude that the performance gains hold across datasets.

3. As the x-axis in Fig.1 is not a hyperparameter or one value that you can change to obtain the y-value, I assume you run multiple experiments to get the curves. Could you also report the standard deviation of the curves, or if that is problematic, maybe a scatter plot?


4. As a minor comment, I think the authors should be more precise when referring to "classifier confidence", and should *not* be mixed with "uncertainty estimation". While I think it is common to mix the two, uncertainty estimation goes beyond the former, as it requires statistical tools to estimate the epistemic and aleatoric uncertainty of the model (for example, population of models). From line 121, I think the authors should use "classifier confidence" where needed (e.g. caption in Fig. 1).


5. Regarding the writing, while I did not include it in my evaluation, I think the authors could improve it in terms of simplifying the sentences, as (in my opinion) some sentences are ambiguous and overloaded, making the manuscript harder to read. For example the sentences in lines 3, 29 can be simplified, and some terms such as "unrobust data", "attackable data" could be made more precise as these depend on the classifier and not solely the data points (e.g. data points whose perturbations fool the classifier into wrong predictions).



--- Minor ---

In figure 1, consider mentioning what variance is referring to.

line 133: as -> with the

line 139: data -> datapoints

inconsistent spacing between lines 157 and 158

--- Recommendation ---

I encourage the authors to benchmark their method against existing methods on corrupted and out-of-distribution datasets. I would be happy to raise my score if the authors could demonstrate that their method is on par with other baselines on those benchmarks.

--- Post-discussion update ---

The authors addressed most of my concerns, thus I increase my score.

**Time Spent Reviewing:**

5

---

> ### Author Response · Authors · 2021-08-10
> **Response to Reviewer jdF6**
>
> Thanks very much for your willingness to raise your score if we can demonstrate our method is on par with other methods on corrupted datasets. We do provide more experimental results (e.g., results of mixup, confidence-calibrated adversarial training (CCAT) [1], BatchEnsemble [2] and Rank-1-BNN [3]) in the answer to Q1 and demonstrate AR-AdaLS is on par with other baselines on the corrupted datasets. We hope that these additional results allay your concerns.
>
> Further, we would like to emphasize that one major contribution of our work is to successfully build a connection between adversarial robustness and calibration. We believe this can greatly impact the research community by inspiring future research at the intersection of different aspects of robustness in machine learning.
>
> We address your concerns below.
> Q1: Benchmark on the corrupted datasets: In the original paper, we report the results of label smoothing-related and ensemble-based methods on CIFAR-10-C and ImageNet-C in Table 2 and use the CIFAR-100 dataset for ablation study in Table 5. Here we merge part of these existing results and compare our AR-AdaLS with four extra existing methods for calibration: mixup, CCAT [1], BatchEnsemble [2] and Rank-1-BNN [3] across all three corrupted datasets: CIFAR-10-C, CIFAR-100-C and ImageNet-C. Here are the results.
>
> Table 1: ECE (%) on the corrupted datasets (Smaller is better).
>
> |Methods  |               CIFAR-10-C | CIFAR-100-C   |ImageNet-C |
> | ------------ | ------------- | ------------- | ------------- |
> |Vanilla    |                        16.7  |        18.2     |         12.8 |
> |LS                 |                 10.1    |       16.3       |       8.2 |
> |Mixup            |                  8.8   |         11.2     |          9.2|
> |CCAT [1]       |                   9.9    |        16.1     |          - |
> |BatchEnsemble [2]    |     12.4    |       14.9      |        8.9|
> |Rank-1-BNN [3]       |       9.0      |       14.2      |        5.9|
> |AR-AdaLS             |          6.4      |      13.2     |         6.8 |
> |AR-AdaLS of Ensemble | 4.4      |         -         |        4.0|
>
> We can see from this table that single AR-AdaLS is better than mixup on CIFAR-10-C and ImageNet-C, better than BatchEnsemble across three datasets, and better than CCAT and Rank-1-BNN on the CIFAR-10-C and CIFAR-100-C. Note that our proposed “AR-AdaLS of Ensemble” can further outperform Rank-1-BNN on ImageNet-C. Based on these results, we can conclude that our proposed AR-AdaLS is on par with the existing baselines for calibration on the corrupted datasets. We will include these extra results into the final version.
>
> Footnote of Table 1: Two missing numbers in the table: 1) Confidence-calibrated adversarial training (CCAT) involves time-consuming adversarial training, therefore we did not run it on ImageNet. 2) “AR-AdaLS of Ensemble” is mainly tested on CIFAR-10 and ImageNet in Table 2 in the original paper and achieves the best calibration performance.
>
> [1] Stutz, David, Matthias Hein, and Bernt Schiele. "Confidence-calibrated adversarial training: Generalizing to unseen attacks." ICML, 2020
>
> [2] Wen, Yeming, Dustin Tran, and Jimmy Ba. "Batchensemble: an alternative approach to efficient ensemble and lifelong learning." ICLR 2020.
>
> [3] Michael W. Dusenberry*, Ghassen Jerfel*, Yeming Wen, Yian Ma, Jasper Snoek, Katherine Heller, Balaji Lakshminarayanan, Dustin Tran. Efficient and Scalable Bayesian Neural Nets with Rank-1 Factors. ICML 2020.
>
> Benchmark on OOD: Considering that our work is mainly to explore the relationship between adversarial robustness and calibration, and further to improve models’ calibration on the clean and shifted data, measuring calibration on out-of-distribution data, which is hard to measure, is not our main focus.  Therefore, in the main paper, we only have a small section (Line 271-277 and Figure 3) showing the extra benefit of our proposed AR-AdaLS: it can significantly reduce the number of low-entropy predictions on OOD data compared to vanilla model and label smoothing. To address your concern with mixup on OOD, we test the the AUCROC score of mixup and our AR-AdaLS using CIFAR-10/CIFAR-100 as in-distribution/OOD data. The AUCROC of mixup is 0.821±0.003 whereas our AR-AdaLS achieves 0.885±0.003. This sufficiently supports that our AR-AdaLS is significantly better than mixup on OOD data.
>
> Q2: Figure 2 is to show how AR-AdaLS outperforms label smoothing by taking CIFAR-10 as an example and we do observe similar patterns on CIFAR-100 and ImageNet. We will take your advice on adding these figures into the final version.
>
> Q3: We can report the standard deviation of the curves and we do observe similar patterns when we run multiple experiments: the data points whose perturbations fool the classifier into wrong predictions are also poorly calibrated.
>
> Q4: Thanks very much for pointing out the inappropriate use of “uncertainty estimation”. We will be more precise when we use “uncertainty estimation” vs “classifier confidence”.
>
> Q5: Thanks very much for the suggestions on simplifying the sentences and rephrasing “unrobust data” and “attackable data” with “data points whose perturbations fool the classifier into wrong predictions”. We will take your suggestions and clarify in the final version.

---

> > ### Comment · Reviewer_jdF6 · 2021-08-27
> > **Thanks for the response and the additional results**
> >
> > I thank the authors for their response and the additional results.
> > Most of my concerns have been addressed, thus, as promised, I am increasing my score.
> > I hope the authors will incorporate the new results and do their best to improve the writing.
> >
> > Thanks

---

### Official Review · Reviewer_fyY9 · 2021-07-19

**Rating:** 5
**Confidence:** 4

**Summary:**

This paper investigates the relationship between calibration and adversarial robustness, showing that calibration error is larger among less robust images. Based on this, a training procedure is proposed where the labels of the images are smoothed based on the robustness level, to produce a better calibrated model.

The proposed method is evaluated on CIFAR-10/CIFAR-100, and their corrupted counterparts, comparing with other calibration and label smoothing approaches, as well as ensemble methods.

**Limitations And Societal Impact:**

The paper does not discuss limitations or potential negative societal impact of their work. This could be improved by identifying the underlying assumptions (e.g. measuring robustness with CW attack) of the model and where they fall short. Or identifying cases of failure and what is their impact.

**Main Review:**

This paper studies an interesting idea: how the less robust images are also responsible for higher calibration error. The idea of ranking the images based on their robustness and exploit the less-robust images to improve calibration is worth investigating.
However there are many issues with the paper in its current version, as listed next:

1. Until the last page of the paper, it was not clear if the robustness levels are pre-computed or update during training. This should be made clear earlier. Considering the "on-the-fly" computation, how does it change over the epochs? Do the images migrate from one level to another? Does the perturbation norm associated with each level change over the epochs? And by how much?
2. How does the proposed methods AdaLS/AR-AdaLS affect the accuracy? In particular, for the comparison with the other approaches in Table 2, it would be useful to compare the accuracy as well, or clearly indicate that there's no significant change in accuracy, if that's the case.
3. Although the authors indicate in the checklist that they present error bars in the plots, none of the plots or tables in the main paper have error bars or standard deviations, even though it is claimed in the paper that results come from 5 independent runs (also, multiple runs with distinct data splits would be more informative than relying on random initialization).
4. Due to the lack of error bars/deviation measurements, it is hard to interpret the significance of the results. For example, in Table 2, Mixup or the ensemble methods have similar or better calibration for CIFAR-100. AR-AdaLS is better only for CIFAR-10 and by a margin of 0.2, which is hard to assess if it is significant.
5. The units used for ECE are inconsistent throughout the paper. In Figures 1 and 2, ECE values are decimals (range from 0 to 1), while in Tables 2, 3, 5 and Figure 4 it is measured in percentages (from 0 to 100).
6. For OOD experiments: CIFAR-10 and CIFAR-100 are derived from the same original dataset and have similar classes. It would be more meaningful to compare with more distinct datasets such as SVHN, LSUN or STL-10.
7. For Figure 3, it could be improved by showing the entropy in-distribution as well. The current figure shows that AR-AdaLS increases entropy OOD, but it's not clear how it affects the in-distribution entropy.
8. Another useful OOD analysis would be to have the ROC plot (or measure AUCROC) for the detection of images in/out of distribution.
9. Notation is a bit confusing or unclear: it is never described if $\tilde{p}_r$ is a vector or a function, or if it is distinct for each example.The $z$ in the superscript is never defined in the text. In the Algorithm, the epoch index should be updated for $t+1$ in the appropriate places in the for loop.

Overall, the paper investigates an interesting idea, but, from the results presented, did not convince me that the proposed method is better than the competing baselines.

**Time Spent Reviewing:**

4 hours

---

> ### Author Response · Authors · 2021-08-10
> **Response to Reviewer fyY9**
>
> Thanks for your time and efforts reviewing our work. Below we address your concerns in detail.
>
> Q1: We will take your advice and move the discussion of robustness levels earlier. For the “on-the-fly” version, we compute the adversarial robustness twice, at epoch 65 and epoch 130 as illustrated in the implementation details in Appendix. Here we compare these two computed adversarial robustness at different epochs. 1) Taking the validation images on CIFAR-10 as an example, the images do migrate from one level to the other. Specifically, there are 32.5% images remaining in the same robustness level; 38% images migrating to the neighboring robustness level. 2) The perturbation norm associated with each level does change over the epochs. Below we list the maximum perturbation norms of each robustness level for epoch 65 and epoch 130.
>
> Epoch 65: 0.040  0.061  0.080  0.099  0.117  0.139  0.167  0.214  0.263
>
> Epoch130:0.048  0.070  0.090  0.109  0.131  0.165  0.203  0.241  0.287
>
> We can see that as the epoch increases, the maximum perturbation norm of each robustness level increases correspondingly. This implies that as the model achieves higher accuracy during training, we need to increase the perturbation to fool the classifier. In addition, the difference of adversarial robustness between epoch 65 and 130 also explains that “on-the-fly” can always provide better calibration performance compared to “pre-computed” because it provides more precise adversarial robustness ranking.
>
> Q2: We have included the accuracy plot in Appendix, see Figure 5. Indeed, AR-AdaLS does not affect the accuracy. We will add in the final version that AR-AdaLS does not lead to any significant accuracy change.
>
> Q3 & Q4: Lack of error bars.  We will add all the error bars in the final version. Note that there are no error bars for ensemble-based methods since each ensemble method is a combination of 5 independent runs. Specifically, to address your concerns about the significance of our method compared to mixup, we want to confirm that AR-AdaLS is significantly better than mixup on CIFAR-10 because the ECE score (%) of AR-AdaLS is 0.6±0.02 while mixup is 0.8±0.06. In addition, AR-AdaLS is also significantly better than mixup on corrupted datasets: CIFAR-10-C and ImageNet-C. The ECE score (%) of mixup on CIFAR-10-C and ImageNet-C is 8.8±0.36 and 9.2±0.20 respectively. In contrast, the ECE score (%) of AR-AdaLS is significantly lower, at 6.4±0.45 and 6.8±0.25 on CIFAR-10-C and ImageNet-C respectively. Taking all these together, we believe that our proposed AR-AdaLS is significantly better than mixup. Please refer to the answer to Q1 to reviewer jdF6 for more experiments that demonstrate our AR-AdaLS is on par with other baselines on corrupted datasets.
>
> Q5: We will modify Figure 1 & 2 to make the units of ECE consistent throughout the paper.
>
> Q6-Q8: OOD experiments. First, we want to emphasize that our work is mainly to explore the relationship between adversarial robustness and calibration and further to improve models’ calibration on the clean and shifted data. Since it is hard to measure the calibration on out-of-distribution data, we only provided the entropy plot as one additional benefit of our model on OOD .
>
> Second, it is common to use CIFAR-100 dataset as OOD data when the model is trained on CIFAR-10, e.g., see [1]. To address the reviewer’s concern, we also plot the entropy using SVHN dataset as OOD data and observe a similar pattern: AR-AdaLS can significantly reduce the number of low-entropy predictions on OOD data compared to the vanilla model and label smoothing.
>
> Third, we follow the work [2] (see Figure 1 (e)) which only plots the entropy on the OOD. To address your concern with in-distribution entropy, we follow your suggestion measuring the AUCROC score: When use CIFAR-10 as in-distribution data and SVHN as OOD: AR-AdaLS achieves 0.937±0.005, which is significantly better than standard label smoothing, whose score is 0.900±0.018. In addition, when using CIFAR-10 as in-distribution data and CIFAR-100 as OOD：the AUCROC score of AR-AdaLS is 0.885±0.003 whereas standard labels smoothing is 0.832±0.005.
>
> In all, we will add these extra results on OOD in the final version. That said, please note that OOD is an extra benefit of our model but not our main focus.
>
> Q9: Notation: We introduce $\tilde{p}_r$ in line 156 which stands for the soft labels for each data point. We will specifically emphasize $\tilde{p}_r$ is a vector in the final version. The $z$ never appears alone in the superscript: either $z=y$ or $z\neq y$, and we have introduced it when it first appears in line 157: $p^{z=y}$ means the label for the correct class and $p^{z\neq y}$ means the labels for the other classes. We will take your advice on using t+1 for the updates in the for loop in the Algorithm.
>
> [1] Liu, Jeremiah Zhe, et al. "Simple and principled uncertainty estimation with deterministic deep learning via distance awareness." NeurIPS 2020.
>
> [2] Snoek, J., Ovadia, Y., Fertig, E., Lakshminarayanan, B., Nowozin, S., Sculley, D., Dillon, J., Ren, 375 J., and Nado, Z. Can you trust your model’s uncertainty? evaluating predictive uncertainty under 376 dataset shift. NeurIPS 2019.

---

> > ### Comment · Reviewer_fyY9 · 2021-08-26
> > **Thank you for your response.**
> >
> > Thank you for your response. While I would still recommend caution when using terms like "significantly better" without conducting hypothesis tests, the inclusion of the standard deviation in the results dispelled some of my concerns of how effective AR-AdaLS is compared to its baselines. The results are still somewhat mixed, as the proposed method is not consistently better in all datasets, but the new results at least show they are on par with other methods such as Mixup.
> >
> > The new results and discussion (including your detailed reply to Q1) adequately addressed most of my concerns, and contribute to an improved paper. As such, I'm increasing my score.

---

> > > ### Author Response · Authors · 2021-09-03
> > > **Further response**
> > >
> > > Thanks for raising your score and we are glad that we adequately addressed most of your concerns.
> > >
> > > We will be more cautious while using ''significantly better '' when comparing single AR-AdaLS with other baselines in the final version. However,  we still want to emphasize that our proposed ``AR-AdaLS of Ensemble'' does perform significantly better than existing baselines, as shown in Table 1 in the response to Reviewer jdF6.
> > >
> > > In addition, one of the main contributions of our work is that we have nicely built a connection between adversarial robustness and calibration: the adversarially unrobust data are more likely to be poorly calibrated. This insight has been greatly appreciated by the other reviewers because it is valuable to the research community who are interested in either adversarial robustness or calibration. We also believe this work can spur further work at the intersection of different research areas related to robustness.

---

### Decision · Program_Chairs · 2021-09-27

**Decision:**

Accept (Poster)

**Comment:**

All reviewers agreed the method proposed in this submission is insightful and novel. The authors' rebuttal has successfully addressed the reviewers' concerns. However, the reviewers are also less satisfied by the fact that the proposed method seems to be, at best, on par with other baselines (e.g. Mixup). The robustness-based label smoothing is not consistently better and there are no theoretical arguments to favor their approach. This is a borderline paper but I recommend acceptance. Despite limited performance gain compared to baseline methods, I believe the technical contributions are sufficient.